# Diagnosis of Periprosthetic Joint Infection: The Utility of Biomarkers in 2023

**DOI:** 10.3390/antibiotics12061054

**Published:** 2023-06-15

**Authors:** Mehmet Kursat Yilmaz, Ahmad Abbaszadeh, Saad Tarabichi, Ibrahim Azboy, Javad Parvizi

**Affiliations:** 1Rothman Orthopaedic Institute, Thomas Jefferson University, Philadelphia, PA 19107, USA; ahmad.abbaszadeh2022@gmail.com (A.A.); saadtarabichi@gmail.com (S.T.); javadparvizi@gmail.com (J.P.); 2Department of Orthopaedics and Traumatology, School of Medicine, Istanbul Medipol University, Istanbul 34810, Turkey; ibrahimazboy@gmail.com

**Keywords:** periprosthetic joint infection, serum, synovial fluid, marker

## Abstract

Periprosthetic joint infection (PJI) is a rare yet devastating complication following total joint arthroplasty (TJA). Early and accurate diagnosis of PJI is paramount in order to maximize the chances of successful treatment. However, we are yet to identify a single “gold standard” test for the diagnosis of PJI. As a result, the diagnosis of PJI is often challenging. Currently, the 2018 ICM definition of PJI is the only validated diagnostic criteria available. This article will review the importance of serum and synovial biomarkers in the diagnosis of PJI. In addition, it will provide a brief overview of the emerging modalities for the identification of infections in this setting.

## 1. Introduction

Periprosthetic joint infection (PJI) is a leading cause of morbidity and mortality in patients undergoing total joint arthroplasty TJA [1]. As the annual volume of primary and revision TJA continues to expand, inevitably, so will the burden of disease secondary to PJI [2,3,4]. Concurrently, it is estimated that the economic burden of this disease process will reach $1.85 billion by 2030 [5].

Regardless of the affected joint, the workup of patients with suspected PJI typically involves a stepwise approach [6]. First, patients undergo a blood draw to have their serological marker levels analyzed. If the serological markers are negative, no further workup is necessary. In cases with elevated serological markers, the next step involves aspiration of the joint and analysis of the synovial biomarkers. Collectively, the results of serum and synovial markers, in conjunction with clinical findings, can help physicians reach a diagnosis of PJI.

A consensus group (400 experts from 52 countries) first convened at the International Consensus Meeting (ICM) held in Philadelphia, Pennsylvania (USA) in 2013 [7]. Although there have been several diagnostic criteria that have been developed for the diagnosis of PJI, in a recent study [6], the 2018 ICM criteria were found to be superior to previously proposed definitions. Using random Forrest analyses, the authors of the 2018 ICM criteria assigned scores to different serum markers, synovial markers, and intraoperative findings. Subsequently, the proposed algorithm underwent external validation and was found to have near-perfect diagnostic utility for the identification of PJI.

The purpose of this review is to examine the utility of serum and synovial biomarkers that have proven utility in the diagnosis of PJI. In addition to this, the present article will also highlight microbial identification and emerging marker technologies in this setting.

## 2. Clinical Features

Periprosthetic joint infection is a devastating complication following TJA. Although a number of symptoms have been shown to be concerning for PJI, patients most commonly complain of a recent history of a painful prosthesis. However, it is important to recognize that there are many causes of pain after joint replacement surgery and that infection is not always the primary cause. Therefore, all patients presenting with a painful joint after TJA should be considered infected until proven otherwise [8].

Early postoperative infections are believed to occur during the implantation of components [9]. Most commonly, symptoms of acute PJI may include pain, swelling, drainage from the wound, surgical site erythema, and effusion [9]. Chronic infections often mimic aseptic failures, but certain symptoms can help differentiate between the two. While aseptic failures may result in weight-bearing pain and limited mobility, persistent pain is most commonly associated with PJI [9,10]. Furthermore, it is also important to recognize that the hematogenous spread from another site is a common cause of chronic PJI [11].

When there is clinical suspicion of infection, it is essential to ask patients about recent illnesses that are suggestive of bacteremia, as well as any history of multiple surgeries on the same joint or previous infections. Factors that increase the risk of skin barrier penetration, such as prolonged periods of wound drainage or delayed wound healing, should also be taken into consideration. Gathering this information can aid in the diagnosis of periprosthetic joint infection and guide the selection of appropriate diagnostic tests [10].

In the diagnostic approach of PJI, imaging techniques such as standard radiographs, ultrasonography, computed tomography, and magnetic resonance imaging can be used [12]. Although these imaging techniques cannot provide consistent information about the presence of infection, they can serve to guide surgeons [12,13]. Nuclear medicine imaging is also used for diagnostic purposes. These are technetium-99m-methyl-diphosphonate scintigraphy, scintigraphy using radiolabeled neutrophils or radiolabeled anti-granulocyte antibodies, and 18F-fluorodeoxyglucose positron emission tomography coupled with CT (FDG-PET/CT) [12]. However, these have not yet broken the trust of many surgeons in biomarkers.

## 3. Diagnostic Tests

When a patient with suspected PJI presents to the clinic, the orthopedic surgeon should not rely on any one single test in order to confirm or refuse a diagnosis of PJI. With the advent of new biomarkers, it can sometimes be challenging to determine which tests are most suitable to help reach a diagnosis of PJI. In an effort to provide clarity and improve overall diagnostic confidence in this setting, the American Academy of Orthopaedic Surgeons (AAOS) introduced evidence-based guidelines for the diagnosis of PJI in 2010 [14]. Subsequently, in 2018, Shohat et al. [6] presented an evidence-based and validated algorithm that offers a stepwise approach to PJI diagnosis. In addition to having near-perfect accuracy in the diagnosis of PJI, the 2018 ICM criteria were found to be superior to the previously proposed definitions (Figure 1). The use of stepwise algorithms can help clinicians make informed decisions before proceeding with more invasive and costly testing, thereby minimizing the number of tests performed at each stage (Figure 2).

### 3.1. Serum and Plasma Markers

Due to their widespread availability, low cost, and high sensitivity, serological inflammatory markers play a crucial role in the workup of patients with suspected PJI [15]. As a result, they are considered the most appropriate initial step in evaluating patients with painful prostheses. However, it is important to note that inflammatory markers are often nonspecifically elevated in patients with certain medical comorbidities. Nevertheless, serological markers can be useful in the initial workup of patients with suspected PJI and should be interpreted in conjunction with other diagnostic tests and clinical findings [16].

#### 3.1.1. Erythrocyte Sedimentation Rate (ESR) and C-Reactive Protein (CRP)

Based on data spanning several years, the erythrocyte sedimentation rate (ESR) and the serum C-reactive protein (CRP) are the most commonly performed serologic tests in the workup of patients with PJI [17]. Due to their high sensitivity, both erythrocyte sedimentation rate (ESR) and C-reactive protein (CRP) levels have been found to be particularly useful in the detection of PJI [18]. However, there is a wide range of reported sensitivity and specificity values for CRP in the literature, varying from 62% to 100% and from 64% to 96%, respectively. Similarly, the reported sensitivity values for ESR range from 33% to 95%, whereas the specificity values range from 60% to 100% [16].

Although CRP and ESR are general parameters that can be influenced by various systemic inflammatory conditions, such as autoimmune disorders, active cancer, coronary heart disease, acute or chronic renal failure, obesity, and infections that are not joint-related, they are still widely used as inflammatory markers to aid in the workup of patients with suspected PJI. Notwithstanding, it is important to note that these markers have poor specificity for the diagnosis of PJI [16].

It is well-recognized that ESR and CRP levels are often normal in cases of PJI caused by low-virulence organisms. Therefore, low levels of these markers cannot be used to exclude PJI [19,20]. At the present time, these non-specific inflammatory markers are commonly utilized in clinical practice as adjuncts to clinical and other laboratory assessments to help make a diagnosis of PJI [18,21].

#### 3.1.2. White Blood Cell (WBC) Count

The serum white blood cell (WBC) count is a commonly used laboratory test that can detect the presence of infection or inflammation. While WBC count alone cannot provide a definitive diagnosis of a particular disease, it is a valuable diagnostic tool in the diagnosis of many diseases [22]. While some studies have shown that serum WBC count can be a useful diagnostic tool for PJI, other studies have reported conflicting results [23,24,25,26]. Toosi et al. [27] conducted a study on the diagnostic accuracy of WBC count for the diagnosis of PJI. The results of their study indicated that the sensitivity and specificity of WBC count for diagnosing PJI were 55% and 66%, respectively. These findings support the widely held notion that serum WBC count and differential have limited utility and should not be part of the workup of patients with suspected PJI.

#### 3.1.3. Polymorphonuclear Leukocyte Percentage (PMN%)

The polymorphonuclear leukocyte percentage (PMN%) is often elevated in a number of disease processes, including acute bacterial infections, neoplasia, and metabolic diseases [28]. Although there are few studies investigating the utility of serum PMN% in the diagnosis of PJI [27,29], it is evident that it has little to no role in the workup of patients with suspected PJI.

#### 3.1.4. Neutrophils to Lymphocytes Ratio (NLR)

The NLR is a cost-effective marker that can be easily calculated from a patient’s routine laboratory work. The relative increase in neutrophil count and decrease in lymphocyte count in bacterial infections suggests that the serum neutrophil-lymphocyte ratio may be useful as a biomarker for the diagnosis of PJI [30]. Numerous investigations have been carried out to evaluate the serum NLR’s sensitivity and specificity in diagnosing PJI. They found that while NLR can be used in the diagnosis of PJI, its utility is limited by its low sensitivity, specificity, and high false-negative rate [16,26]. According to Yombi et al. [31] and Zhao et al. [32], NLR may be a promising marker for the detection of acute PJI as its levels tend to normalize faster than CRP in patients undergoing TJA.

Notwithstanding, the literature on the diagnostic utility of the neutrophil-to-lymphocyte ratio (NLR) for PJI is still in its early stages, and further studies with larger sample sizes are necessary in order to determine its value in this setting.

#### 3.1.5. Platelet Count to Mean Platelet Volume Ratio (PC/mPV)

Due to advancements in automated counters, it is now possible to measure the number and size of platelets in an efficient manner [33]. In one study, the authors found that an increase in platelet count and a decrease in mean platelet volume were associated with active infection [34]. More recently, there have been studies examining the utility of serum platelet count and platelet size ratio in the diagnosis of PJI [35,36,37,38]. In a study by Paziuk et al. [39], the authors evaluated a large cohort of revision TJA patients and found that at a cutoff of 31.7%, PC/mPV demonstrated a sensitivity and specificity of 48% and 81%, respectively, for the diagnosis of PJI. Despite initial enthusiasm for its use, the low sensitivity and overall accuracy of PC/mPV prevented it from being universally adopted in the workup of patients with suspected PJI.

#### 3.1.6. Procalcitonin

In contrast to its utility for systemic bacterial infection, serum procalcitonin has been shown to have limited utility in the diagnosis of PJI [40,41,42,43,44,45]. In a study by Busch et al. [46], the authors examined the role of serum PCT in the diagnosis of PJI and found it to have a very low sensitivity (13%). Subsequently, a recent meta-analysis by Yoon et al. concluded that serum PCT had little to no role in the workup of patients with suspected PJI.

#### 3.1.7. D-Dimer

D-dimer is a substance produced during the breakdown of blood clots that can be detected in both whole blood and plasma and is often used as a marker for fibrin formation and degradation [47]. D-dimer testing is an important screening test for venous thromboembolisms, specifically deep-vein thrombosis and pulmonary embolism [48]. D-dimer has more recently attracted attention as a potential marker for infection in patients with bacteremia and sepsis [49]. Subsequently, several studies [50,51,52,53,54] have been conducted in order to determine the role of D-Dimer in the diagnosis of PJI, culminating in its inclusion in the 2018 ICM criteria [17].

In a recent prospective study, Tarabichi et al. [55] found that D-dimer was not inferior to CRP and ESR in diagnosing PJI, with a sensitivity and specificity of 81% and 82%, respectively. Furthermore, they found that D-dimer demonstrated the highest sensitivity for PJI caused by indolent organisms at 94%.

However, it is important to exercise caution when interpreting D-dimer levels in patients with inflammatory and hypercoagulable disorders as diagnostic thresholds may be higher in this patient population [56].

#### 3.1.8. Fibrinogen

Fibrinogen plays an important role not only in hemostasis and wound healing but also serves as a key player in antimicrobial host defense against bacterial infection via the utilization of various mechanisms [57]. Subsequently, its potential as a biomarker for the diagnosis of infection has been demonstrated in the literature.

In a systematic review and meta-analysis [58], the authors found that plasma fibrinogen had a sensitivity and specificity of 79% and 73%, respectively. These results were found to be lower than those of serum D-dimer. Nevertheless, according to the study by Xu et al. [59], plasma fibrinogen has demonstrated a high level of sensitivity (93.8%) and specificity (77.4%) in detecting PJI before revision TJA, indicating its potential as a valuable diagnostic marker in this setting. Moreover, when combined with CRP, the specificity of this biomarker was improved to 93.5%. Several similar studies have also highlighted the significance of plasma fibrinogen in the diagnosis of PJI [25,35,60,61].

Although fibrinogen can aid in the diagnosis of PJI and its comparable performance to serum CRP has been demonstrated previously, it may not be sufficient to definitively confirm or exclude PJI. Hence, fibrinogen could be used as an adjunct test in the workup of patients with suspected PJI.

#### 3.1.9. Interleukin-6 (IL-6)

IL-6 is a cytokine that is produced in response to infections and tissue injuries. It also plays a significant role in host defense by stimulating acute phase responses, hematopoiesis, and immune responses. Moreover, it promotes the differentiation of B lymphocytes, which activates T lymphocytes and regulates the synthesis of acute phase proteins such as C-reactive protein (CRP) and fibrinogen. Consequently, IL-6 can be utilized as a diagnostic marker for PJI [62,63].

In a recent systematic review and meta-analysis by Li et al. [64], which encompassed thirty studies, the pooled sensitivity and specificity of serum IL-6 in detecting PJI ranged from 0.76 to 0.88. Interestingly, the literature reports varying sensitivity and specificity values at different cutoff values when investigating the utility of IL-6 in detecting PJI. Therefore, currently, relying solely on serum IL-6 levels for the diagnosis of PJI appears to be challenging. Further studies are necessary to ascertain the precise value of IL-6 [42,65,66,67,68].

#### 3.1.10. Albumin/Globulin Ratio

General serum markers, such as total protein, albumin, and total globulin, have been extensively studied in order to determine their utility in the assessment of immunological and nutritional status in patients with malignant, inflammatory, and infectious diseases. In everyday practice, these markers are often employed either individually or in combination with CRP to provide valuable insights [69,70].

In the most recently published study by Wang et al. [71], it was found that the albumin/globulin ratio exhibits greater diagnostic efficacy compared to ESR and demonstrates comparable diagnostic performance to CRP. This study revealed that the sensitivity and specificity of the albumin/globulin ratio were 91.07% and 73.58%, respectively, whereas CRP had a sensitivity and specificity of 78.57% and 88.68%, respectively, in the diagnosis of PJI. In a study conducted by Zhang et al. [72], the authors found that the albumin/globulin ratio has the potential to be a valuable diagnostic biomarker for PJI, as it produces similar results to commonly used biomarkers, such as ESR and CRP.

However, further studies are necessary in order to determine the clinical utility of the albumin/globulin ratio as a diagnostic biomarker for PJI, as current evidence suggests that its value may be limited.

#### 3.1.11. IL-1β and TNF-α

Interleukin (IL)-1β, a powerful inflammatory protein, is primarily synthesized by activated immune cells (monocytes, microglia, and macrophages) [73]. Erdemli et al. and Gollwitzer et al., in their studies in which they examined many markers, could not find a significant place in the diagnosis of serum IL-1β [74,75]. In a study by Bottner et al. [40], the sensitivity and specificity of TNF-α were found to be very low. Thus, both need stronger evidence to be used as diagnostic markers.

### 3.2. Synovial Markers

In recent decades, synovial fluid analysis has emerged as one of the more useful diagnostic tools in the workup of patients with suspected PJI [76,77]. Joint aspiration not only provides an opportunity for the evaluation of different markers but also allows for cytological and microbiological examinations [77]. Although the search for a definitive synovial fluid test continues, several potential biomarkers have been identified [78].

#### 3.2.1. WBC Count and PMN Percentage

The synovial fluid white blood cell count (WBC) and polymorph nuclear leukocyte percentage (PMN%) have been recognized as highly accurate, consistent, and widely-available tests [79]. According to the International Consensus Meeting (ICM) definition in 2018 [17], an elevated synovial WBC count (>3000 cells/μL) corresponded to 3 points, whereas an elevated PMN% (>80%) corresponded to 2 points.

In a retrospective study conducted by Diniz et al. [80], which involved 102 patients who underwent revision THA or TKA, five synovial parameters were examined in order to determine their utility in the diagnosis of PJI. They found that synovial WBC count had the highest utility for the diagnosis of PJI, with an area under the curve (AUC) of 0.94. It was followed closely by PMN%, with an AUC of 0.91. Furthermore, in a systematic review by Qu et al. [81], it was concluded that synovial fluid WBC count and PMN% are sufficient and possess a clinically acceptable diagnostic value in identifying infection following THA and TKA.

However, there are certain factors that need to be taken into consideration when evaluating the results of synovial fluid biomarkers. The use of antibiotics before joint aspiration is a common practice and has been associated with lower WBC counts and PMN% [82]. The presence of blood mixed with synovial fluid can complicate both quantitative and qualitative examination [83]. Different patient-specific factors, such as the presence of frank pus, inflammatory arthritis, metal-on-metal arthroplasty, crystal-induced arthritis, and small-volume aspirations, can also have an impact on the accuracy of synovial cytology results [84,85].

#### 3.2.2. Synovial CRP

Routinely, in clinical practice, the CRP level is commonly elevated during inflammatory states. In recent years, synovial CRP has also been measured from synovial fluid samples in patients with suspected PJI. Despite initial promising reports, more recent studies have found that synovial CRP provides no additional benefit when compared to serum CRP in the diagnosis of PJI [80,86,87,88,89,90,91,92,93].

Parvizi et al. [91] conducted a diagnostic study where synovial biomarkers were compared to serum equivalents in the diagnosis of PJI. In this study, synovial CRP was included for the first time and demonstrated superior performance compared to serum equivalents. Additionally, Baker et al. [89] conducted a study involving the largest cohort of patients undergoing both serum and synovial CRP analyses, following the latest validated criteria for PJI. In this study, synovial CRP exhibited an AUC of 0.951, with a sensitivity of 74.2% and a specificity of 98%. These results were superior to those of serum CRP, indicating a higher diagnostic performance of synovial CRP.

In light of these proven studies and evidence, elevated synovial CRP has been accepted as one of the latest criteria by the 2018 ICM for the diagnosis of PJI [17]. This recognition emphasizes the significance of synovial CRP as a useful diagnostic adjunct in the workup of patients with suspected PJI.

#### 3.2.3. Alpha-Defensin

In recent years, the synovial fluid alpha-defensin test has gained widespread popularity as a diagnostic tool and has been included in the 2018 ICM as one of the minor criteria [17]. Two methods have emerged for detecting synovial fluid alpha-defensin in recent years: the enzyme-linked immunosorbent assay (ELISA) and the lateral flow device. ELISA is a laboratory-based assay that provides results within 24 h, whereas the lateral flow device rapidly detects infection within 20 min without requiring laboratory testing. However, pooled results from various studies have indicated that the synovial fluid alpha-defensin ELISA method exhibits higher sensitivity compared to the lateral flow test [94,95,96,97]. According to a recent meta-analysis, synovial fluid alpha-defensin has been shown to have the highest sensitivity for the diagnosis of PJI [98]. This non-microbiological test is considered a reliable reference for intraoperative microbiological diagnosis [98]. However, it is worth noting that some studies have argued against the routine use of alpha-defensin due to its nonsuperior diagnostic accuracy and higher cost when compared to conventional synovial marker factors. According to the results of a study by Shohat et al. [99], alpha defensin was found to provide no additional benefit over leukocyte esterase in the diagnosis of PJI. Additionally, in another study, the authors concluded that the diagnostic accuracy of synovial alpha defensin for the diagnosis of PJI was comparable and not statistically superior to the combination of synovial WBC count and PMN% [100].

However, it is important to note that alpha-defensin may have perceived advantages when compared to other synovial fluid markers for the diagnosis of PJI. One notable advantage is that alpha-defensin appears to maintain its diagnostic accuracy even in the presence of antibiotic treatment [101]. Additionally, it remains accurate even when the synovial fluid is contaminated with blood [102]. Finally, it exhibits consistent sensitivity in detecting a wide spectrum of pathogens, including organisms with low virulence [103]. This broad pathogen detection capability enhances its diagnostic utility in identifying infections caused by various types of pathogens.

Despite the advantages of synovial alpha-defensin testing, its widespread inclusion in routine tests faces challenges due to its high cost and limited accessibility. Consequently, the integration of synovial alpha-defensin as a routine diagnostic marker for PJI remains a challenge.

#### 3.2.4. Leukocyte Esterase (LE)

The alteration in the color of the LE strip is caused by the reaction between the leukocyte esterase, which is secreted by leukocytes, and the substrate present on the test strip [104]. The activity of leukocyte esterase is typically considered to be directly proportional to the number of leukocytes [105,106].

The LE test strip is widely used as a biomarker in synovial fluid for PJI due to its affordability, convenience, and commercial availability [107]. However, a significant challenge associated with this test strip is the absence of a well-defined cutoff value. Since it relies on colorimetric analysis, it is susceptible to various factors that can significantly affect its results, such as the presence of blood [108]. To address the issue of blood interference, synovial fluid samples should be centrifuged before being applied to the LE strip [109]. Furthermore, it should be noted that the LE strip test has a high false-negative rate when a cutoff of 2+ is used.

In 2017, Shahi et al. [110] performed a significant study that examined the test performance of LE urine strip tests. The study analyzed a total of 659 synovial specimens, making it the largest study cohort in the current literature. The authors reported a sensitivity and specificity of 75% and 90.9%, respectively, in the diagnosis of PJI.

Despite the disadvantages associated with the leukocyte esterase test, it is considered a minor criterion in the 2018 ICM definition of PJI.

#### 3.2.5. IL-6

IL-6 is recognized as an important cytokine that induces inflammation under septic conditions, and its concentration is significantly increased in septic patients [63]. In a study by Catterall et al. [111], they proposed that when serum markers are elevated systemically, they may also be elevated in the synovial fluid. Therefore, the examination of these markers in the synovial may provide useful diagnostic information. Synovial markers may gain value in this regard. Thus, several researchers have conducted research to examine the potential value of synovial IL-6 in the diagnosis of PJI. While some studies have suggested that IL-6 may be of some use in the workup of patients with suspected PJI [88,112,113], others have questioned its diagnostic utility in this setting [79,114].

The heterogeneity in the sample measurement of synovial IL-6 among these studies has led to a spectrum of proposed diagnostic thresholds. This indicates that synovial IL-6 is not yet a reliable biomarker for the diagnosis of PJI.

#### 3.2.6. Calprotectin

Calprotectin is an innate immune protein that is primarily produced and released by neutrophils and macrophages in response to inflammatory stimuli [115]. It is released during an inflammatory response as a defense against bacterial infections. It also holds the potential to serve as an autonomous diagnostic biomarker for preoperative PJI due to its remarkable sensitivity (95%) and specificity (95%) [116].

While further research on calprotectin is necessary, according to recent diagnostic meta-analyses [117,118] synovial calprotectin has the potential to be a promising biomarker for diagnosing PJI due to its benefits, including its high diagnostic accuracy and convenience. However, it is important to note that current data are insufficient in order to be able to consider synovial calprotectin as a suitable stand-alone diagnostic marker for PJI.

#### 3.2.7. IL-1β and TNF-α

Deirmengian et al. [112] found that synovial fluid IL-1β levels were more accurate in diagnosing infection than ESR, CRP level, or synovial fluid WBC count. IL-1β levels in the synovial fluid were 258 times higher and TNF-α levels were 4 times higher in the infected group than in the aseptic group. However, in a recent study, they found that synovial fluid IL-1β was more valuable than ESR and CRP in the diagnosis of PJI, and its sensitivity was 95% with 95% specificity at a 95% confidence interval [119]. Synovial fluid TNF-α was also found to be unsuitable for diagnosis [120]. However, further studies are needed to use these markers in the diagnosis of PJI.

### 3.3. Microbial Identification

In general, staphylococci account for more than 50% of PJI cases. In addition, polymicrobial cases are not uncommon. Culture negative PJI should also be kept in mind [121].

#### 3.3.1. Culture

Regarding the role of culture, a study by Michael et al. concluded that in order to maximize the diagnostic yield of traditional culture, at least five samples should be taken intraoperatively and held for a minimum of 8 days [122]. In another study conducted by K. Keely Boyle et al., they mentioned that aspiration culture has reasonable sensitivity and specificity compared to tissue culture, and patients can be treated according to aspiration culture results [123]. Notwithstanding, tissue culture for the diagnosis of underlying polymicrobial PJI should be performed [124].

#### 3.3.2. Molecular Technologies

PCR

We are yet to determine where PCR can be useful as a method of pathogen detection in patients with PJI. In a recent meta-analysis, the authors found that PCR results can be reliable and of great value in the diagnosis of PJI with a sensitivity of 75% and a specificity of 96% [125]. In another study, it was concluded that despite the fact that multiplex-PCR could be a useful adjunct for pathogen identification, it is not superior to conventional culture [126].

b.NGS

Metagenomic next-generation sequencing (mNGS) as a method of pathogen detection is increasingly becoming more readily available in the diagnosis of PJI. In a recent study, the authors found that mNGS can enhance the diagnosis rate of PJI, especially when used in conjunction with the culture [127]. However, it is important to note that the false-positive rate of this technology is still quite high and remains a significant concern in this setting [128].

c.Matrix-assisted laser desorption ionization time-of-flight mass spectrometry (MALDI-TOF MS)

The primary benefit of MALDI-TOF technology in bacterial identification lies in the rapidity of obtaining results, with reports indicating a decrease in processing time from 24–48 h to under one hour when conducting routine identification of bacterial colonies cultivated on specified agar [129,130]. In a recent study [131], direct MALDI-TOF MS analysis of positive BCB-SF showed improved diagnostic capabilities in detecting PJI when used with conventional sonication fluid culture, and this combined approach significantly reduced the time.

#### 3.3.3. Emerging Technologies and Novel Biomarkers

Although there are numerous established markers routinely used in the diagnosis of PJI, ongoing studies are rapidly identifying novel markers and technologies. Keemu et al. [132] examined the success of synovial fluid and serum cytokine and chemokine patterns in the diagnosis of PJI. Their findings suggested that these patterns may be valuable diagnostic tools for the identification of acute PJI. Furthermore, they found that synovial fluid sTNF-R2 demonstrated the highest diagnostic utility for the identification of PJI. Similarly, the performance of new technologies and innovative approaches in the preoperative diagnosis of PJI has also been examined. The success of mass spectrometry has been investigated and has shown promise in providing accurate results [133]. Jacovides et al. [113] conducted a study involving 74 patients, comparing molecular markers between septic and aseptic TJA groups. They observed different levels of sensitivity and specificity in the diagnosis of PJI based on the levels of VEGF, CRP, α2-macroglobulin, IL-8, and IL-6 in the synovial fluid.

In the diagnosis of PJI, many markers (such as IL6, IL17, neopterin, presepsin, BAFF, TNF-R2, osteocalcin, and CD64) that trigger a complex inflammatory response such as proinflammatory cytokines are focused on [74,120,132,133,134,135,136].

## 4. Conclusions

PJI after total hip or knee arthroplasty is a serious complication that can significantly affect a patient’s physical, emotional, social, and financial well-being [137,138]. Therefore, achieving an early and accurate diagnosis of PJI is paramount in order to maximize the chances of successful treatment. The diagnosis of PJI involves a comprehensive approach that combines clinical assessment, serologic testing, synovial fluid aspiration, radiographic evaluation, and microbiologic and histopathologic finding [17]. While some orthopedic surgeons and researchers have attempted to identify a single gold standard test for PJI diagnosis, such a test has not yet been established. Currently, the most up-to-date and comprehensive resource is the 2018 ICM criteria [17].

In this review, we specifically focused on the role that current standard-of-care serum and synovial biomarkers play in the workup of patients with suspected PJI. However, it is worth noting that there are several novel markers that have not yet been universally adopted. Achieving an easy and early diagnosis of PJI holds the potential to reduce the morbidity and mortality secondary to this disease process. Nonetheless, future research should attempt to identify a single marker that can definitively rule out or diagnose PJI.

## Figures and Tables

**Figure 1 antibiotics-12-01054-f001:**
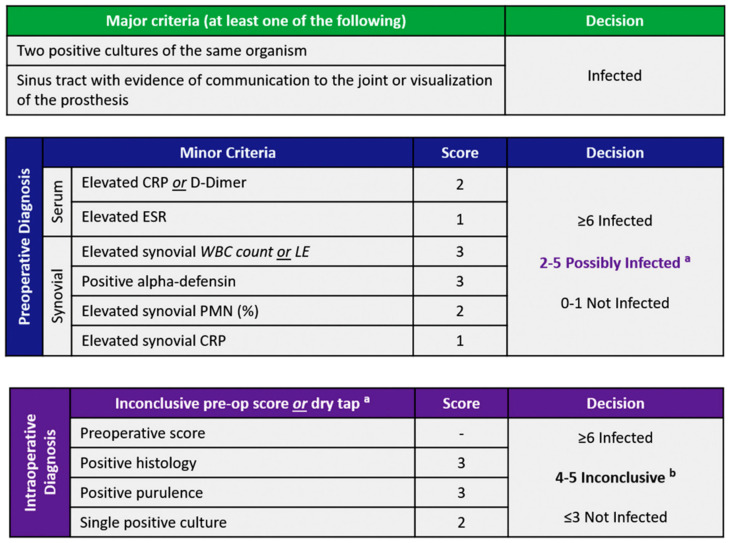
2018 İnternational Consensus Meeting criteria for the diagnosis of PJI. CRP, C-reactive protein; ESR, erythrocyte sedimentation rate; LE, leukocyte esterase; PMN, polymorphonuclear; WBC, white blood cell. ^a^: For patients with inconclusive minorcriteria, operative criteria can also be used to fulfill definition for PJI. ^b^: Consider further molecular diagnostics such as next-generation sequencing.

**Figure 2 antibiotics-12-01054-f002:**
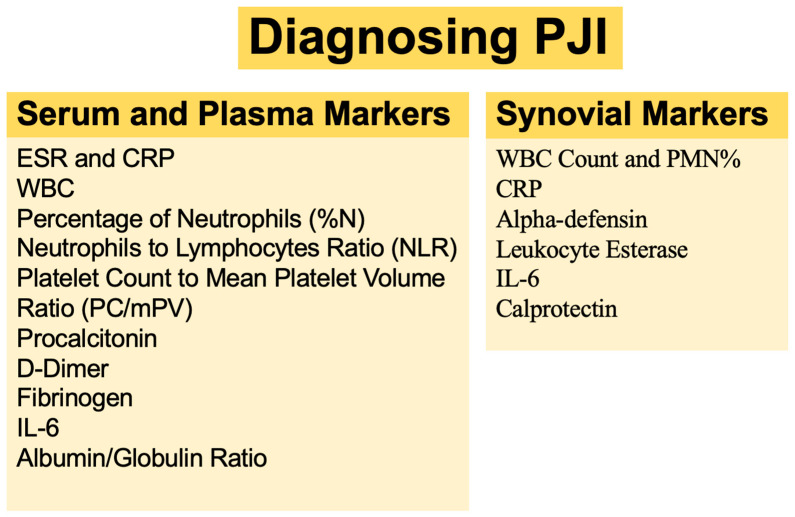
Commonly used serum-plasma and synovial markers for diagnosing PJI.

## Data Availability

No new data were generated in this study. It does not contain confidentiality and ethical restrictions.

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
