# Peer review of "Diagnosis of Periprosthetic Joint Infection: The Utility of Biomarkers in 2023"

_antibiotics, 2023, doi:10.3390/antibiotics12061054_

Round 1

Reviewer 1 Report

The overall logic of the article is clear and innovative, but there are some shortcomings that need to be revised in time.

Manuscript: antibiotics-2439942

The manuscript is entitledDiagnosis of Periprosthetic Joint Infection: Where Are We Now?

In this study, Periprosthetic joint infection (PJI) is a rare yet devastating complication following total joint arthroplasty (TJA). Early and accurate diagnosis of PJI is paramount in order to maximize the chances of treatment success. However, we are a yet to identify a single “gold standard” test for the diagnosis of PJI. As a result, the diagnosis of PJI is often challenging. Currently, the 2018 ICM definition of PJI is the only validated diagnostic criteria available. This article will review the importance of serum and synovial biomarkers in the diagnosis of PJI. In addition, it will provide a brief overview of emerging modalities for the identification of infection in this setting. However, we think there are still some issues in this manuscript need to be corrected by the author before publication.

1.  In the conclusions, the author say that it is worth nothing that there are several novel markers that have not yet been universally adopted. But he did not specify what the new markers were.

2.  In the summary of each marker, most of the authors are the conclusions of predecessors and lack of personal introduction.

3.  with a sensitivity of 74,2% and a specificity of 98%.”, Is there a mistake on line 251 and 252.75.2% of Wei should be modified.

The article is smooth overall, without many grammatical errors.

Author Response

Thank you very much for your review.

We updated the project with your suggestions.

1- We added and touched on Emerging technologies and Novel Biomarkers.

2- As this is a review, we have tried to avoid personal opinions as much as possible.

3- We fixed typos

Thank you

Reviewer 2 Report

Thanks for your invitation to review this manuscript. 

This manuscript reviewed most of the serum and indicators in the diagnosis of PJI. I just have several suggestions: 

1. the role of IL-1β and TNF-α in diagnosis of PJI should be reviewed. 

2. The author should review more indicators which was found in recently years instead of traditional indicators. (PMID: 34056503, PMID: 34555458, PMID: 29305046, et. al) 

3. Serum calcium level can be a useful indicator in the diagnosis of osteomyelitis. The author should discuss if serum calcium can be a good indicator of PJI.  

Language quality is good

Author Response

Thank you very much for your review.

We updated the project with your suggestions.

1-2-3- We touched on this and recent articles more in Emerging technologies and Novel Biomarkers.

Reviewer 3 Report

Dear Authors 

I have carefully reviewed your manuscript "Diagnosis of Periprosthetic Joint Infection: Where Are We Now?" and have the following queries/suggestions:

1. Line 18: The introduction of PJI is very brief and needs to be expanded. 

2.  Authors should explain the factor affecting PJI in clinical features to explain the complexity of PJIs. 

3. What is ICM? Expand on the initial text of the manuscript.

4. Authors should present a tabular form of diagnostic tests with the type of test, sample volume, sensitivity, turnover time, advantages, disadvantages, etc. 

5. Before microbial identification, please discuss the prominent pathogens in PJIs. 

6. Can we use MALDI-TOF/RT-PCR for the identification of pathogens? The authors should discuss it in the microbial identification section. 

7. Typos and grammar need to be checked thoroughly—for example, lines 174, 190. Use 0.00 format over 0,00 in presenting numerical data. 

Thanks 

The quality of English is good but needs to be checked for typos. 

Author Response

Thank you very much for your review.

We updated the project with your suggestions.

Thank you

Reviewer 4 Report

This a well-written review of the state-of-the-art diagnosis criteria for Periprosthetic Joint Infection (PJI). The authors went through a list of serum and synovial biomarkers and discussed their importance/sensitivity/specificity in the diagnosis of PJI. Overall, the manuscript provides a comprehensive overview of the current approaches to diagnosing PJI, while also highlighting the need for further research to optimize diagnostic strategies. I expect it will be of interest to the readers of Antibiotics and to the researchers in the field of bone & joint infections.

Below are my suggested revisions, all of which are minor and/or aesthetic.

Suggested revisions:

1.     Lines 101 and 102: WBC should be the abbreviation of “white blood cell”, not “white blood cell count”.

2.     Line 114: it’s either missing a word between the two commas or having an extra comma.

3.     Line 290: it’s better to put the subsection title “Leukocyte Esterase (LE)” on the next page together with the relevant paragraphs.

Author Response

(The authors gave the same response as above.)

Round 2

Reviewer 2 Report

The authors didn't address my questions clearly and carefully. More indicators I mentioned before must be reviewed one by one in both Circulation and Synovial parts. Serum calcium levels in the diagnosis of PJI should be discussed in the manuscript. 

Author Response

Dear reviewer

First of all, thank you for your review.

1- We added IL-1ß and TNFα captions to both serum and synovial markers.

2- However, while there is a study on the serum calcium level in chronic osteomyelitis, it has no place in the diagnosis of PJI yet. That's why we think it's better not to mention it.

Thank you.